# Removal of Cationic Dyes by Iron Modified Silica/Polyurethane Composite: Kinetic, Isotherm and Thermodynamic Analyses, and Regeneration via Advanced Oxidation Process

**DOI:** 10.3390/polym14245416

**Published:** 2022-12-10

**Authors:** Afiqah Ahmad, Siti Nurul Ain Md. Jamil, Thomas S. Y. Choong, Abdul Halim Abdullah, Nur Hana Faujan, Abel A. Adeyi, Rusli Daik, Nurhanisah Othman

**Affiliations:** 1Department of Chemistry, Faculty of Science, Universiti Putra Malaysia, Serdang 43400, Malaysia; 2Centre of Foundation Studies for Agricultural Science, Universiti Putra Malaysia, Serdang 43400, Malaysia; 3Department of Chemical and Environmental Engineering, Faculty of Engineering, Universiti Putra Malaysia, Serdang 43400, Malaysia; 4Institute of Nanoscience and Nanotechnology, Universiti Putra Malaysia, Serdang 43400, Malaysia; 5Department of Chemical and Petroleum Engineering, College of Engineering, Afe Babalola University Ado-Ekiti, ABUAD, KM. 8.5, Afe Babalola Way, PMB 5454, Ado-Ekiti 360101, Nigeria; 6Department of Chemical Sciences, Faculty of Science and Technology, Universiti Kebangsaan Malaysia, Bangi 43600, Malaysia

**Keywords:** methylene blue, malachite green, polyurethane, iron modified silica, adsorption, kinetic, isotherm, thermodynamic, regeneration, advanced oxidation process

## Abstract

Emerging dye pollution from textile industrial effluents is becoming more challenging for researchers worldwide. The contamination of water by dye effluents affects the living organisms in an ecosystem. Methylene blue (MB) and malachite green (MG) are soluble dyes with a high colour intensity even at low concentration and are hazardous to living organisms. The adsorption method is used in most wastewater plants for the removal of organic pollutants as it is cost-effective, has a high adsorption capacity, and good mechanical stabilities. In this study, a composite adsorbent was prepared by impregnating iron modified silica (FMS) onto polyurethane (PU) foam to produce an iron modified silica/polyurethane (FMS/PU) composite. The composite adsorbent was utilised in batch adsorption of the cationic dyes MB and MG. The effect of adsorption parameters such as the adsorbent load, pH, initial dye concentration, and contact time were discussed. Adsorption kinetics and isotherm were implemented to understand the adsorption mechanism for both dyes. It was found that the adsorption of MB and MG followed the pseudo-second order model. The Langmuir model showed a better fit than the Freundlich model for the adsorption of MB and MG, indicating that the adsorption occurred via the monolayer adsorption system. The maximum adsorption capacity of the FMS/PU obtained for MB was 31.7 mg/g, while for MG, it was 34.3 mg/g. The thermodynamic study revealed that the adsorption of MB and MG were exothermic and spontaneous at room temperature. In addition, the regeneration of FMS/PU was conducted to investigate the composite efficiency in adsorbing dyes for several cycles. The results showed that the FMS/PU composite could be regenerated up to four times when the regeneration efficiency dropped drastically to less than 20.0%. The impregnation of FMS onto PU foam also minimised the adsorbent loss into the environment.

## 1. Introduction

Dyes are widely used in textile, leather, plastic, cosmetic, and food industries. The wide utilisation of dyes has resulted in the uncontrolled disposal of dye effluents into the water system and has raised a major concern among researchers. Methylene blue (MB) and malachite green (MG) are some of the cationic dyes used for dyeing cottons and leather, and gives colour to foods. MB has also made its way in terms of its application in the medical fields, for example, as a therapeutic agent in cardiac surgery and critical care [1]. Meanwhile, MG has been reported to be applied illegally in aquaculture production as an antiprotozoal and antifungal medication for fish [2]. MB and MG are soluble in water but their presence could have a negative impact on aquatic life by blocking the sunlight from passing through water due to their high colour intensity at low concentration [3]. In addition, the ingestion of MB may result in gastrointestinal irritation; lips, mouth, and throat irritation and drowsiness for humans [4]. MG has also been reported to have potential in being teratogenic or carcinogenic to human [5]. The chemical structure of MB and MG are illustrated in Figure 1 [6,7]. By considering all factors discussed, these cationic dyes are deemed suitable as a model to be evaluated in this study.

The adsorption of the solute using inorganic adsorbents is widely applied due to the chemical and thermal stabilities as well as the ability of the adsorbents to be regenerated [8]. The adsorption of organic waste is highly dependent on the adsorbent properties: porosity, high surface area, and good mechanical and thermal stability. These properties can be easily found on silica-based materials. Popular silica-based adsorbents include silica beads, alunite, perlite, dolomite, and zeolite [8]. Silica-based adsorbents have been reported to be efficient in the removal of a wide range of dyes in a batch system including anionic [9], reactive, and cationic [10]. A silica–polymer hybrid (H_HPMA_) synthesised by Jamwal et al. was more effective and selective in MB adsorption compared to MG and Congo red, with a maximum adsorption capacity of 56.6 mg/g [10]. Mobarak et al. reported the maximum adsorption capacity of 75.5 mg/g for cobalt ferrite silica nanocomposite, with the highest MG removal (>90.0%) at pH 6 [11]. In another study, Zirak et al. found that increasing the adsorbent dosage of the carboxymethyl cellulose magnetic nanoparticles (CMC-Fe_3_O_4_@SiO_2_) led to a decrease in MB removal due to adsorbent aggregation. The study limited the adsorbent dosage to 0.03 g and obtained the maximum adsorption capacity of 22.7 mg/g [12].

Adsorbent regeneration is deemed important due to its benefits to the environment and economy in terms of the elimination of secondary pollution and cost effectiveness. Various regeneration methods have been reported such as thermal, chemical, and biological treatment. Advanced oxidation process (AOP) is an example of chemical treatment, which comprises ozonation, photocatalysis, electrochemical oxidation, and the Fenton process [13]. The Fenton reaction involves a rapid generation of hydroxyl radicals, which oxidise and degrade organic pollutants. The regeneration of adsorbents by AOP has been reported to excel in maintaining the adsorbent adsorption performance. In a previous study, iron modified bentonite reported by Gao et al. maintained an adsorption capacity of 168.8 mg/g after five cycles of Rhodamine B adsorption. The regeneration was the optimum at pH 3 under visible light [14]. Kaolinite was used in the adsorption of Rhodamine B, regenerated by AOP, and the removal of dye was slightly reduced to 86.0% after five cycles [15]. Recent work by Shen et al. found that powdered activated coke was regenerated by AOP to remove the chemical oxygen demand (COD) from biological effluent. The regeneration efficiency of the adsorbent reduced to 65.6% only after six cycles [16]. Based on these findings, it is expected that modification of silica with iron could produce a recyclable adsorbent with high adsorption capacity and recyclability. The presence of iron in the silica adsorbent can be used to react with the oxidant to produce radicals for adsorbed dye degradation.

The utilisation of adsorbent support is important to facilitate the recovery of adsorbent particles from the adsorption system, hence preventing the adsorbent loss and avoiding the formation of secondary pollution in water. Highly porous polyurethane (PU) foam is a potent support for small adsorbent particles as it provides high surface area [17]. In addition, porous PU benefits any process that requires high void volumes, breathability, and uniform structure for performance efficiency [18]. Recent reports have proven that PU can also be used as a carrier for microorganisms [17], catalysts [19], and adsorbents [20]. To the best of the authors’ knowledge, no study has been reported on the adsorption of cationic dyes using FMS supported on PU foam. In addition, very few reports on the regeneration by using the AOP of the supported adsorbent have been found.

Previously, we have reported the synthesis and characterisation of FMS/PU [21]. In this paper, FMS/PU composite was utilised for the removal of cationic dyes from an aqueous solution. Silica was chosen in this work because of its high adsorption capacity and stability in a wide range of pH. Iron was incorporated onto silica to assist the adsorbent regeneration process. Iron acted as a catalyst for the decomposition of hydrogen peroxide to produce highly reactive hydroxyl radicals. These radicals degraded the adsorbed dyes on the surface of the silica, hence regenerating the composite for the next cycle of adsorption. AOP was used for the regeneration process instead of desorption because of its efficiency to degrade organic pollutants into less harmful, low-molecular weight compounds, water, and carbon dioxide. In addition, the AOP method eliminates the formation of secondary pollution during the regeneration process. PU foam was selected as a support to assist the recovery of the FMS adsorbent from water. Implementation of the support is expected to eliminate the tedious filtering process and reduce the loss of the adsorbent during the recovery process. The data from the batch adsorption process were further evaluated by using kinetic, adsorption isotherm models and thermodynamic study including pseudo-first and pseudo-second order, Elovich, intraparticle diffusion, Langmuir, and Freundlich to have a better understanding of dye adsorption. To the best of the authors’ knowledge, the utilisation of a novel FMS/PU composite adsorbent in cationic dye batch adsorption has not yet been reported elsewhere. The novel combination of an adsorbent–catalyst and a carrier is expected to improve the currently used method in the water treatment industry. This work is anticipated to contribute to the development of cationic dye adsorption through a sustainable and environmentally friendly supported adsorbent and its regeneration.

## 2. Materials and Methods

### 2.1. Materials

Palm oil polyol (OHV = 220 mg of KOH/g) was obtained from the Malaysian Palm Oil Board (MPOB) (Selangor, Malaysia). Tolylene-2,4-diisocyanate (85.0%) (TDI), polypropylene glycol 1200, tetraethyl orthosilicate 98.0% (TEOS), iron (III) perchlorate hydrate (Fe(ClO_4_)_3_․H_2_O), malachite green (MG), and methylene blue (MB) were purchased from Sigma Aldrich (Gillingham, UK). N,N-dimethylcyclohexylamine and polysiloxane 1280 were purchased from Merck (Darmstadt, Germany) and Dow Corning (Torrance, CA, USA), respectively. Ethanol 95.0%, sodium hydroxide, hydrochloric acid, hydrogen peroxide (H_2_O_2_), potassium nitrate (KNO_3_), and ammonia solution were purchased from R&M Chemicals (Selangor, Malaysia).

### 2.2. Preparation of Iron Modified Silica/Polyurethane (FMS/PU) Composite

The synthesis of iron modified silica (FMS) and flexible polyurethane (PU) foam and the impregnation of FMS onto PU were carried out as reported by Ahmad et al. [21]. FMS was prepared by using sol–gel method. Briefly, TEOS and Fe(ClO_4_)_3_․H_2_O were dissolved in ethanol and heated to 80 °C. Then, the sol–gel formation was initiated by the addition of ammonia solution. The FMS powder obtained was dried at 100 °C and calcined at 400 °C. PU foams were prepared by the one-pot method, where palm oil-based polyol was mixed with isocyanate in the presence of additives. The preparation of the FMS/PU composite was carried out by adding a known mass of FMS into a beaker containing 95% ethanol. The suspension was sonicated for 10 min. A known mass of PU foam with a dimension of 2.0 cm × 2.0 cm × 1.0 cm was immersed and gently stirred in the suspension. The resulting composite was dried at 60 °C for 45 min. Immersion and drying processes were repeated four times to ensure the maximum impregnation of the adsorbent. The final product was washed with distilled water to remove excess FMS and dried at 60 °C for 24 h [21].

### 2.3. Characterisation of FMS/PU Composite

High-resolution transmission electron microscopy (HRTEM) micrographs were captured on a JEOL JEM 2100F Field Emission TEM (Tokyo, Japan). X-ray photoelectron spectra were measured using Auger electron spectroscopy with an X-ray photoelectron spectrometer (AES-XPS) Kratos/Shimadzu Axis Ultra DLD (Manchester, England). The X-ray diffraction (XRD) pattern of the regenerated FMS/PU was examined using a PANalytical X’Pert PRO (Malvern, UK) with an operating voltage of 30 kV and current of 30 mA. Fourier transform infrared (FTIR) spectra were recorded by using a BX Perkin Elmer (Akron, OH, USA) with the universal attenuated total reflectance (UATR) technique to identify functional groups that are present in the spent FMS/PU.

### 2.4. Point of Zero Charge (PZC)

The point zero charge was used to determine the point where the surface charge is zero. The solid addition method was used to determine the point zero charge of the FMS/PU adsorbent. FMS/PU (0.1 g) was added into each of eight conical flasks containing 50 mL of KNO_3_ (0.01 M) solution. The initial pH of the KNO_3_ solutions was adjusted to pH 3, 4, 5, 6, 7, 8, 9, and 10 using KOH (0.1 N) and HNO_3_ (0.1 M) solutions. After 48 h of agitation, all conical flasks were withdrawn from the shaker and the final pH was recorded. The intersection point on the curve ∆pH (pH_i_–pH_f_) versus pH_i_ was considered as the value of the point zero charge.

### 2.5. Batch Adsorption Experiments

The adsorption of dyes (MB and MG) was carried out in batch to study the effect of various parameters: adsorbent dose, pH, reaction time, and initial concentration. The experiments were carried out by varying one parameter at a time. FMS/PU composites with different adsorbent dosages (0.100 g–0.225 g) were placed in 100 mL of dye solutions and agitated in a thermostatic water bath shaker at 100 rpm. The initial pH of the dyes was adjusted using KOH (0.1 M) and HCl (0.1 M) at different pH values of 3, 5, 7, 9, and 11. The effect of temperature was investigated by varying the temperature of the reaction system in the range of 25–55 °C. The effect of the initial dye concentration was carried out by varying the dye concentration in the range of 20–100 mg/L. The effect of contact time for different concentrations was determined by taking out the sample at specified intervals for 200 min. The composite was then separated from the reaction and the concentrations of MB and MG were determined using a Shimadzu UV-1280 spectrophotometer (Kyoto, Japan) at the 663.8 nm and 617.0 nm wavelengths, respectively.

The percentage of dye removal, R, and the adsorption capacity at the equilibrium, *Q_e_* (mg/g) were calculated using Equations (1) and (2), respectively:(1)$$\mathrm{Percentage}\;\mathrm{removal}\;\left(\%\right)=\frac{\left(C_{0}-C_{e}\right)}{C_{0}}\times 100\%$$
(2)$$\mathrm{Adsorption}\;\mathrm{capacity}\;\left(Q_{e}\right)=\frac{\left(C_{0}-C_{e}\right)V}{m}$$
where *C_0_* is the initial concentration of dyes (mg/L); *C_e_* is the equilibrium concentration of dyes (mg/L); *V* is the volume of the reaction solution (L); and *m* is the mass of FMS loaded (g).

### 2.6. Regeneration of FMS/PU

The regeneration of FMS/PU was carried out by the advanced oxidation process (AOP) to degrade the adsorbed dyes. For the adsorption process, the FMS/PU composite was placed in a flask containing 80 mg/L of dye and was agitated for 3 h. Meanwhile, 1000 mg/L of hydrogen peroxide solution was added into 100 mL of distilled water in a different flask. The dye saturated FMS/PU was then immersed in hydrogen peroxide solution and agitated for 20 h. The regenerated FMS/PU was recovered and reused for the next cycle of dye adsorption. The regeneration process continued until the regeneration efficiency was reduced to 50.0% or less. The regeneration efficiency was calculated by comparing the adsorption capacity of fresh and regenerated FMS/PU via the following Equation (3):(3)$$\mathrm{Regeneration}\;\mathrm{efficiency}\;\left(R_{e}\right)=\frac{Q_{R}}{Q_{F}}\times 100$$
where *Q_F_* and *Q_R_* are the adsorption capacities of the fresh and regenerated FMS/PU, respectively.

## 3. Results and Discussions

### 3.1. Characterisation of FMS/PU Composite Adsorbent

The HRTEM micrographs of FMS powder at different magnifications are shown in Figure 2. Dark areas contributed by the iron oxide particles were not evenly scattered throughout the matrix. According to the micrographs, the iron oxide particles were irregular with non-uniform particle size and shape. This finding is in agreement with the broad XRD pattern, elemental analysis, and previous reports; the FMS powder adsorbent exhibited an amorphous characteristic [22,23,24].

The XPS analysis was conducted to determine the elemental composition and the chemical state of the FMS/PU composite. In Figure 3, a wide spectrum of FMS/PU and narrow scan of C 1s, O 1s, N 1s, Si 2p and Fe 2p are presented. The wide spectrum of FMS/PU confirmed the presence of iron, silicon, and oxygen; the main components of FMS powder. Carbon and nitrogen were mainly contributed by PU foam. In Figure 3b, the peaks centred at binding energies of 284.4 and 286.0 eV were assigned to the C-C/C=C aromatic ring and C-O (alkoxy), respectively [25]. Meanwhile, in Figure 3c, the appearance of peaks at 532.9 eV and 534.3 eV could be resolved to C=O and Si-O, respectively [26]. The narrow scan for N 1s in Figure 3d yielded a single peak at 396.4 eV, which was assigned to the amide groups [25]. Figure 3e presents the Si 2p spectrum consisting of two peaks with binding energies of 101.9 eV (O-Si-O) and 103.1 eV (Si(O)_4_) [26]. The XPS high-resolution scan on Fe 2p (Figure 3f) showed a single peak at 726.8 eV, which corresponded to Fe^3+^ [27]. Taking into consideration the Fe 2p binding energy and the absence of the chlorine peak in the wide scan, it was concluded that the iron precursor used in the sol–gel synthesis was fully reacted and existed in an oxide form, Fe_2_O_3_. A similar result was observed in a detailed XPS study on a silica-based material containing iron oxide [28]. The presence of the single peak in the N 1s spectrum and the absence of Si–C or Si–O–C peaks in the XPS analysis confirmed that PU foam and the FMS adsorbent formed a physical interaction. This finding is in line with the FTIR results reported in the authors’ previous publication; the absence of an additional functional group in FMS/PU compared to PU indicates that the interaction between the FMS powder and PU foam was physical interaction. A detailed characterisation and explanation on the preparation of the FMS/PU composite were discussed by the authors in a previous publication [21]. 

### 3.2. Batch Adsorption

#### 3.2.1. Effect of Adsorbent Load

The adsorbent load was varied to determine the optimum FMS load in the removal of MB and MG. Figure 4 shows the percentage removal of both dyes at different FMS loadings (0.100 g, 0.125 g, 0.150 g, 0.175 g, 0.200 g and 0.225 g). The graphs show an increase in the percentage removal of dyes as the FMS load increased. The percentage of MB removed gradually increased up to 87.0% and reached equilibrium when the FMS load was 0.200 g. A similar trend was observed in the adsorption of MG, where the dye removal gradually increased up to 91.2% and achieved a constant dye removal when the FMS load was 0.200 g. The gradual increase in the dye removal was the result of the increase in available adsorption sites. The percentage of dye removal achieved equilibrium as the adsorption was limited by the amount of available adsorbate during the adsorption process. On the other hand, the adsorption capacity for both dyes decreased as the FMS load increased. This can be explained by the increase in unoccupied adsorption sites as the amount of dye molecules to be adsorbed was kept constant, resulting in a decrease in the adsorption capacity. The optimum FMS load of 0.200 g was further used in the following studies of MB and MG adsorption.

#### 3.2.2. Effect of Initial pH

The initial dye pH is deemed important as dye adsorptions are often affected by the solution pH. The effect of pH on the removal of MB and MG is represented in Figure 5. It was observed that MB removal increased sharply from pH 3 (46.3%) to pH 5 (85.1%) and achieved the highest removal at pH 9 (90.3%), before a slight drop at pH 11 (89.1%). Meanwhile, the removal of MG was the highest at pH 7 (98.6%) and decreased at pH 11 (47.4%). Based on these results, it was concluded that the adsorption of MB and MG by FMS/PU is highly dependent on the initial dye pH. The adsorption of MB and MG increased with increase in pH due to the composite adsorbent surface charge and the charge of thee dyes. The pH of the point zero charge (pH_PZC_) for FMS/PU was determined to be pH 4, as shown in Figure 6. Thus, the surface of FMS/PU was positively charged at pH < pH_PZC_ and negatively charged at pH > pH_PZC_. The pK_a_ value for MB and MG was 3.8 and 6.9, respectively [29,30,31,32]. The MB and MG dye molecules exist in their cationic form when pH > pK_a_ [33,34]. Hence, increasing the pH of the dye solutions resulted in the increase in the electrostatic attraction of positively charged dye molecules towards the negatively charged surface of the FMS/PU composite. A higher pH was also reported in a previous study to have a better removal of MB and MG for the silica-based adsorbent [30,33,35,36,37,38]. Therefore, for further adsorption study, pH 9 and 7 were selected as the optimum pH for MB and MG adsorption, respectively.

#### 3.2.3. Effect of Solution Temperature

Adsorption temperature plays an important role in the adsorption process. The change in the adsorption capacity as a function of temperature may indicate the spontaneity of the adsorption process and the type of interaction between the adsorbent and adsorbate [39]. Figure 7 illustrates the effect of the adsorption temperature on the percentage removal of dyes and the number of adsorbed dyes. At room temperature (25 °C), 90.4% of MB removal was observed with the highest adsorption capacity of 8.96 mg/g. The percentage removal decreased to 43.8% at 35 °C, followed by 29.3% and 19.5% at 45 and 55 °C, respectively. The trend for MG adsorption was similar to MB, with the percentage removal of 96.0% at room temperature, followed by 51.6%, 43.9%, and 15.7% at subsequent temperatures. It was clear that the percentage of MB removed and the adsorption capacity of FMS/PU decreased significantly as the temperature increased. These results indicate that the dye adsorption onto FMS/PU was favoured at room temperature. The same results were observed in previous reports for the adsorption of MB and MG by silica-based materials [40,41].

#### 3.2.4. Effect of Initial Dye Concentration

The MB and MG initial concentrations were varied to investigate the maximum adsorption capacity of the FMS/PU composite. The effect of the initial dye concentration on the removal percentage of MB and MG are shown in Figure 8. The percentage of dye removal decreased with an increase in the dye concentration. As the dye concentration increased, the number of available sites for adsorption became limited, thus decreasing the dye removal before reaching equilibrium. Meanwhile, the adsorption capacities of both dyes were proportional to the dye concentration. The increasing trend in the adsorption capacities indicated a high adsorption amount of dye at lower concentrations due to excessive active sites of the FMS/PU available. The maximum adsorption capacity obtained for MB and MG were 28.3 mg/g and 38.0 mg/g, respectively. As the adsorption capacity of FMS/PU beyond 80 mg/L remained at equilibrium, that concentration was regarded as the optimum for both MB and MG. The concentration was used in a regeneration study of the FMS/PU composite for the adsorption of MB and MG. 

#### 3.2.5. Effect of Contact Time

The adsorption of dyes is highly dependent on the contact time of the adsorbent with dye molecules [42]. Hence, the effect of contact time was carried out to determine the adequate period for composite adsorbent utilisation. Figure 9a,b represent the effect of the contact time of FMS/PU on the removal of MB and MG at different concentrations, respectively. It was observed that the rate of MB adsorbed increased rapidly in the first 12 min of contact time. While for MG, the rapid adsorption was observed in the first 15 min. After the rapid adsorption period, the rate of MB and MG adsorbed slightly decreased until a constant dye removal was reached. The constant value of dye removal occurred after about 160 min and 140 min for MB and MG, respectively. The constant values represent an equilibrium of the dye concentration that had been adsorbed. The rate of adsorption can be explained as follows. The concentrations of dyes were high at the beginning of the adsorption process, with high vacant active sites on the adsorbent surface. This led to a faster rate of adsorption in the first 12 min and 15 min for MB and MG, respectively. Afterwards, the adsorption sites of the adsorbent became saturated; hence, reduced the rate of dye adsorption onto the surface binding sites. At 160 min and 120 min, the adsorption of MB and MG achieved the equilibrium state, respectively. At that state, no more dye was able to be adsorbed as the binding sites were fully occupied and saturated. 

### 3.3. Kinetic Studies

The adsorption kinetics of MB and MG were evaluated at different dye concentrations of 20, 40, 60, 80, and 100 ppm, at a constant FMS load of 0.200 g and pH of 9 and 7, respectively. Pseudo-first order, pseudo-second order, and Elovich kinetic models were applied in order to describe the dye adsorption rate. The intraparticle diffusion model was also employed to delineate the diffusion roles on the dye sequestration mechanism in the adsorption process. The equations of the pseudo-first order (4), pseudo-second order (5), Elovich model (6), and intraparticle diffusion (7) are expressed below, respectively:(4)$$\log\left(Q_{e}-Q_{t}\right)=\frac{-k_{1}t}{2.303}+\log Q_{e}$$
(5)$$\frac{t}{Q_{t}}=\frac{t}{Q_{e}}+\frac{1}{k_{2}Q^{2}{}_{e}}$$
(6)$$Q_{t}=\frac{1}{\beta }\ln\left(\alpha \beta \right)+\frac{1}{\beta }\ln\left(t\right)$$
(7)$$Q_{t}=k_{IP}t^{0.5}+C_{IP}$$
where *Q_e_* and *Q_t_* are the adsorption capacity (mg/g) at equilibrium and at time *t*, respectively. *k*_1_ is the rate constant of the pseudo-second first adsorption, *k*_2_ is the rate constant of the pseudo-second order adsorption (min^−1^). *α* is the Elovich sorption rate constant and *β* is the Elovich constant, which correspond to the extent of surface coverage. *k_IP_* is the rate constant for intraparticle diffusion and *C_IP_* is the boundary layer diffusion effect. 

The calculated parameters of four different kinetic models for MB and MG are shown in Table 1 and Table 2, respectively. The pseudo-second order model showed better agreement between the *Q_e_* calculated and *Q_e_* experimental for both MB and MG compared to the pseudo-first order model. In addition, the correlation coefficient, *R*^2^, of the pseudo-second order model was higher for MB and MG compared to the pseudo-first order model. This indicates that the adsorption kinetics of MB and MG dyes onto the FMS/PU followed the pseudo-second order model. The pseudo-second order model assumes that the rate limiting step is chemisorption, where the adsorption involved the sharing or exchange of electrons between the adsorbent and adsorbate. This is in agreement with previous reports on MB and MG adsorption by silica-based adsorbents [10,35,43].

The Elovich kinetic model showed a good fit with the experimental data based on the high correlation coefficient, *R*^2^ values (>0.9300), as stated in Table 1, indicating that the model is acceptable for the description of MB and MG dye entrapment by the FMS/PU composite adsorbent. The low values of *α* and *β* confirmed the Elovich model viability that the desorption rate was lower than the dye adsorption rate [44]. It also implies that chemisorption is a dominant force in the sorption of cationic dyes and the sequestration process occurred at the heterogeneous surfaces of the FMS/PU [45,46].

Intraparticle diffusion was employed to determine the rate limiting step of the adsorption process (graph is not shown here). The positive values of *C_IP_* for both MB and MG dyes in Table 2 indicate a fast initial sorption process and surface adsorption on the FMS/PU composite [47]. The linear line did not pass through the origin and this implies that intraparticle diffusion was not the only rate limiting step for both MB and MG uptake by FMS/PU [46,48,49]. Thus, the combination of intraparticle diffusion and chemical adsorption played a vital role in the cationic dye entrapment onto the FMS/PU surface.

### 3.4. Adsorption Isotherms

Adsorption isotherms were carried out to study the interaction between the solutes and the adsorbent, and the distribution of the solute between the solid and the liquid phase. Adsorption studies were carried out by varying the dye concentrations from 20 to 100 mg/L at room temperature. The Langmuir and Freundlich isotherms model were selected in this study to better understand the adsorption behaviour. The Langmuir isotherm defines the sorption as homogenous, where dye molecules adsorbed onto the surface of the adsorbent have equal sorption energy [50]. On the other hand, the Freundlich isotherm defines a heterogenous system but is not restricted to monolayer formation. The linear forms of the isotherm equations are as follows:

Langmuir equation:(8)$$\frac{1}{Q_{e}}=\frac{1}{k_{L}C_{e}Q_{max}}+\frac{1}{Q_{max}}$$
where *C_e_* is the equilibrium concentration of the dye; *Q_max_* is the maximum monolayer adsorption capacity; and *k_L_* is the Langmuir constant. 

The separation factor, *R_L_*, was calculated in order to determine the favourability of the adsorption process for both MB and MG using the equation:(9)$$R_{L}=\frac{1}{1+K_{L}C_{0}}$$
where *C_0_* is the initial concentration of the dye solution. 

Freundlich equation:(10)$$\log Q_{e}=\frac{1}{n}\log C_{e}+\log k_{F}$$
where *C_e_* is the equilibrium concentration of dye, and the Freundlich constants *k_F_* and *n*, where *k_F_* is the adsorption capacity of FMS/PU and 1/*n* indicates heterogeneity of the adsorbent surface and the intensity of adsorption. The heterogeneity of the adsorbent surface increased as the value of 1/*n* was closer to zero. The adsorption was deemed favourable when 0 < 1/*n* < 1, unfavourable when 1/*n* > 1, and irreversible when 1/*n* = 1 [51]. 

The values obtained from the Langmuir and Freundlich models are presented in Table 3. Based on the correlation coefficient, *R*^2^ values of higher than 0.9000, both the Langmuir and Freundlich isotherms showed a good fit to the experimental data. However, the Langmuir model showed a better fit to describe the adsorption process for MB and MG due to the closest value of the calculated *Q_max_* and experimental *Q_max_*. Hence, on top of the kinetic analyses result, the cationic dye sequestration process at the adsorbent surface might also occur via the monolayer adsorption system [52,53].

The separation factor, *R_L_*, indicates the nature of the adsorption, where *R_L_* = 0, irreversible; 0 < *R_L_* < 1, favourable; *R_L_* = 1, linear; and *R_L_* > 1, unfavourable [54]. Favourable cationic dye uptake by FMS/PU was observed for both MB and MG according to the values of separation factor, *R_L_*, were between 0 and 1. The monolayer adsorption capacity, *Q_max_*, calculated for MB and MG were 31.7 mg/g and 34.3 mg/g, respectively. Table 4 shows a comparison of *Q_max_* for several types of silica-based adsorbents in the respective dye adsorption. The *Q_max_* of FMS/PU was comparatively higher than previous reports (Table 4) for other silica-based adsorbents regarding the entrapment of both cationic dyes.

### 3.5. Thermodynamic Studies

In order to have a better understanding of the adsorption mechanism, a thermodynamic study was carried out to determine the Gibbs free energy, ΔG° (kJ/mol), enthalpy change, ΔH° (kJ/mol), and entropy change, ΔS° (J/mol K). The thermodynamic parameters were calculated using Equation (11):(11)$$K_{d}=\frac{Q_{e}}{C_{e}}$$
(12)$$\ln K_{d}=\frac{\Delta S{}^{\circ}}{R}-\frac{\Delta H{}^{\circ}}{RT}$$
(13)ΔG°=−RTlnb
where *K_d_* is the equilibrium constant; *C_e_* and *Q_e_* are the equilibrium concentration and adsorption capacity, respectively. *R* is the universal gas constant (8.314 J mol^−1^ K^−1^) and *T* represents the absolute temperature (K). The values of ΔH° and ΔS° were obtained from the slope and the intercept of ln *K_d_* against 1/*T*. 

Table 5 summarises the thermodynamic parameters of MB and MG adsorption onto FMS/PU. The negative value of ΔH° also suggests that the adsorption process was exothermic with the probability of having physical adsorption [39]. A decrease in disorder and randomness at the solid–liquid interface during the adsorption of dyes onto the FMS/PU surface was proven by the negative values of ΔS° in both the MB and MG adsorption [40]. The negative values of ΔG° at room temperature for both MB and MG adsorption indicated that the adsorption was spontaneous and favoured at low temperature rather than high temperature [49]. The positive values of ΔG° at higher temperature also implied that the rate of desorption was higher than the rate of adsorption, resulting in a lower percentage of dye removal at higher temperature [46]. 

### 3.6. Adsorption Mechanism

The functional groups in FMS/PU and the spent composites were determined using FTIR analysis. Figure 10 shows the FTIR spectra of FMS/PU, the spent adsorbent after the adsorption of MB (FMS/PU—MB), and spent adsorbent after the adsorption of MG (FMS/PU—MG). There was no additional peak observed in the FMS/PU—MB and FMS/PU—MG. The intensity of N–H stretching (3290 cm^−1^) and Si–O–Si symmetric vibration (796 cm^−1^) for FMS/PU—MB and FMS/PU—MG were higher compared to the fresh FMS/PU. This can be due to the interaction of the adsorbed dyes and the composite. 

Based on the results obtained and discussed, the mechanism of MB and MG adsorption was probably a combination of chemical and physical adsorptions. Electron sharing or transfer may occur between the FMS/PU composite and cationic dyes. The physical adsorption process can be attributed to the electrostatic interaction and hydrogen bonding between FMS/PU and the cationic dyes. Figure 11a,b illustrates the mechanism of the adsorption of MB and MG, respectively. The electrostatic attraction was considered as the major contributor in the adsorption process, which highly depends on the reaction pH. The adsorption of MB and MG were favoured at high pH value (pH 9 and pH 7, respectively) and at low temperature. The MB and MG dye molecules were positively charged at the specified reaction pH, while the surface of FMS/PU was negatively charged [57]. The large number of the negatively charged oxygen group induced the fast adsorption of positively charged dye molecules. Hydrogen bonding between dye molecules and the FMS/PU also played a part that further supports the cationic dye adsorption process [59]. Hence, increasing the pH of dye solutions resulted in the increase in the electrostatic attraction of positively charged dye molecules towards the negatively charged surface of the FMS/PU composite. 

### 3.7. Regeneration of Adsorbent

Dye saturated FMS/PU was regenerated by using an advanced oxidation process (AOP) that degraded the adsorbed dyes. FMS/PU was initially agitated in 80 mg/L of dyes for 3 h, at the optimum dye pH. The composite was then transferred into H_2_O_2_ solution for the regeneration process. The regenerated FMS/PU was then reused for the next adsorption cycle. The percentage of MB and MG removal by FMS/PU and regeneration efficiency for each cycle are presented in Figure 12. The cycle refers to the regeneration cycle, where cycle zero represents the first adsorption before the FMS/PU was regenerated. Based on Figure 10, FMS/PU was regenerated three times in MB adsorption before the regeneration efficiency dropped drastically to 18.53% while the regeneration efficiency of FMS/PU in MG adsorption gradually decreased until cycle three and then dropped to 12.26% on the cycle four. The drastic decrease might be attributed to the iron leached out, as previously reported [21]. The iron content in the composite decreased over the cycles, resulting in a reduction in hydroxyl radicals being produced during the regeneration process. Hence, the percentage of dye removal dropped drastically due to the cumulative decrease in degradation activity. A previous study showed that a silica-based adsorbent can be regenerated via AOP for three to five cycles without losing its efficiency in dye adsorption [15,62,63]. However, the concentration of dyes used in the adsorption–regeneration process in previous reports were up to 30 mg/L, which was lower than the concentration used in this work, of 80 mg/L. Hence, it was concluded that FMS/PU has potential in dye treatments with a recyclability of up to two cycles for MB and three cycles for MG. 

XRD analysis was carried out on spent FMS/PU to observe any change in the crystallinity of the composite. Figure 13 shows a comparison of the XRD patterns of FMS/PU, FMS/PU—MB, and FMS/PU—MG. A significant change was not observed in any XRD patterns due to the amorphous state of FMS/PU. Hence, the XRD pattern proves that the FMS/PU retains its stability after the adsorption of MB and MG.

### 3.8. Future Direction

In a typical batch process, wastewater is mixed with the adsorbent in a predetermined condition (reaction time, pH, temperature) and the final adsorbate concentration is measured at the end of the adsorption process. Regeneration of the adsorbent is often conducted via sedimentation, filtration, or centrifugation [64]. The typical process of the batch removal of the pollutant is illustrated in Figure 14. The typical batch process is time consuming and costly. However, a simplified method was conducted in the present work in which the FMS adsorbent powder was impregnated onto the surface of the PU foam carrier. By implementing the carrier for the adsorbent, the adsorbent recovery was way easier, eliminating the need for the sedimentation and filtration process. As discussed in the regeneration of the FMS/PU section, the regeneration of the adsorbent was eased and a minimal loss of adsorbent was observed. Since the impregnation of FMS powder onto the PU surface was achieved via physical interaction, there was a possibility of FMS leached from the carrier. This situation may lead to a high total suspended solid (TSS) in the reaction system. Despite the absence of a chemical bond between FMS and PU, our previous report found that the TSS in the treated dye effluent was in the acceptable range of TSS according to the Malaysian Environmental Quality (Industrial Effluent) Regulations 2009. Hence, this composite is deemed to be stable and did not contribute towards secondary pollution. In addition, the FMS/PU composite is highly potent for application in continuous adsorption, which is widely used in water treatment plants. The implementation of a carrier has been reported to improve the adsorption capacity of the adsorbent [65]. In future work, FMS/PU may be utilised in a fixed-bed column method to have a better understanding of the composite performance for the continuous adsorption of cationic dyes. 

## 4. Conclusions

In the present work, FMS/PU was prepared and utilised in the adsorption of the cationic dyes MB and MG. The plausible physical interaction between the FMS powder and PU surface was discussed. The effects of the adsorbent load, initial pH, reaction temperature, initial dye concentration, and contact time on the removal of dyes were investigated. It was found that the percentage removal of dyes increased when the adsorbent load increased with the optimum adsorbent load of 0.200 g for both MB and MG. The percentage removal was the highest at pH 9 for MB and pH 7 for MG. The removal of MB and MG by FMS/PU favoured a low reaction temperature. An increase in the initial dye concentration resulted in a reduction in the dye percentage removal and an increase in the adsorption capacity of FMS/PU. A longer contact time between the FMS/PU adsorbent and dye molecules resulted in a higher percentage of dye removal before achieving equilibrium. The adsorption kinetics of MB and MG dyes onto the FMS/PU was best described by the pseudo-second order model, indicating a chemisorption process. The Langmuir model fitted the adsorption of the MB and MG data, indicating that the adsorption occurred via a monolayer adsorption system. The maximum monolayer adsorption capacity of FMS/PU obtained for MB adsorption was 31.7 mg/g, while for MG, it was 34.3 mg/g. The thermodynamic study revealed that the adsorption of MB and MG were exothermic and spontaneous at room temperature. The FMS/PU composite could adsorb concentrated dyes and be regenerated up to three times for MB and four times for MG before losing its adsorption efficiency. Based on the results discussed in this paper, the FMS/PU composite adsorbent has a promising potential for cationic dye adsorption. The sustainable and environmentally friendly adsorbent has a comparably high adsorption capacity and high recyclability, which makes it beneficial for industry and the environment. 

## Figures and Tables

**Figure 1 polymers-14-05416-f001:**
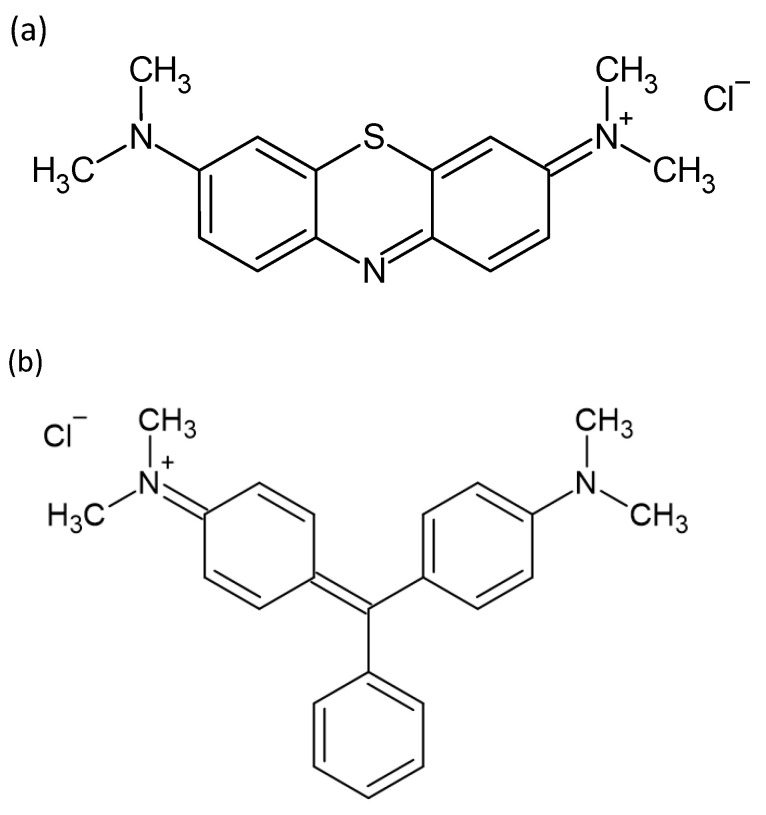
Chemical structure of (**a**) methylene blue (MB) and (**b**) malachite green (MG) [6,7].

**Figure 2 polymers-14-05416-f002:**
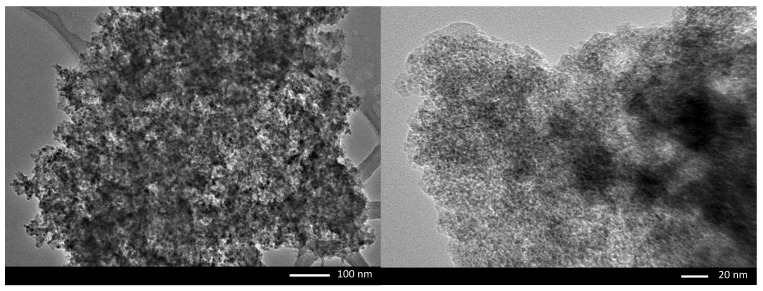
The HRTEM micrographs of the FMS powder adsorbent.

**Figure 3 polymers-14-05416-f003:**
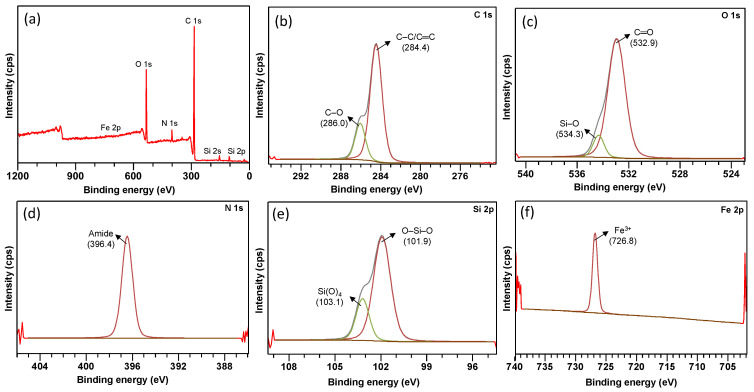
(**a**) XPS survey spectrum and high-resolution spectra of (**b**) C 1s, (**c**) O 1s, (**d**) N 1s, (**e**) Si 2p, and (**f**) Fe 2p of FMS/PU composite adsorbent.

**Figure 4 polymers-14-05416-f004:**
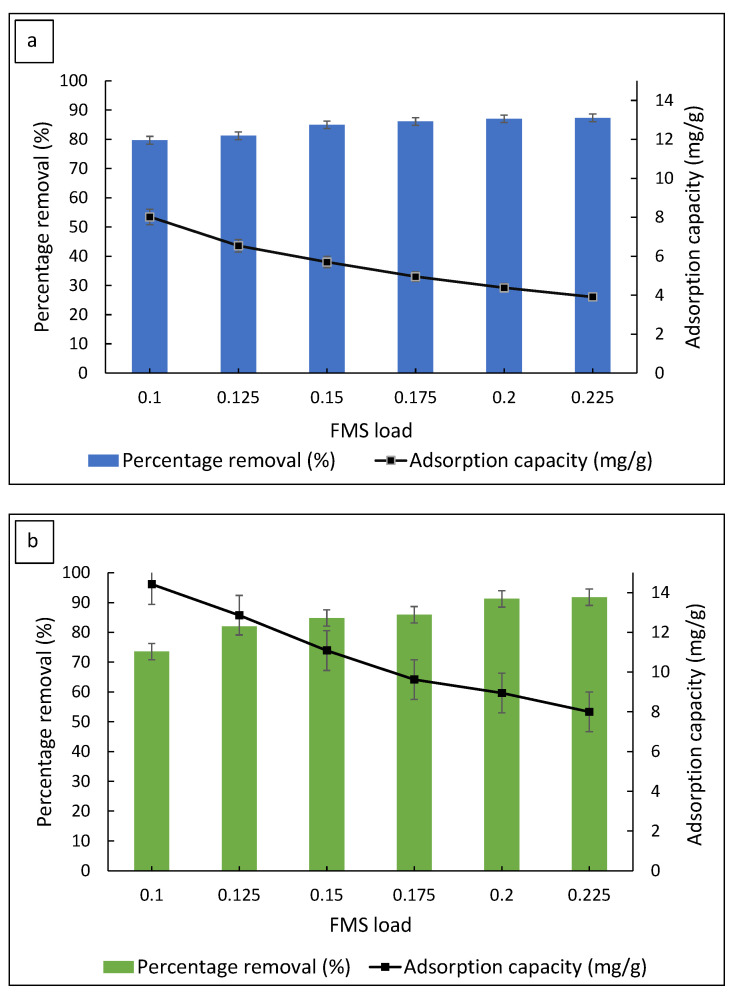
The effect of the FMS/PU adsorbent load on the percentage removal of (**a**) MB and (**b**) MG.

**Figure 5 polymers-14-05416-f005:**
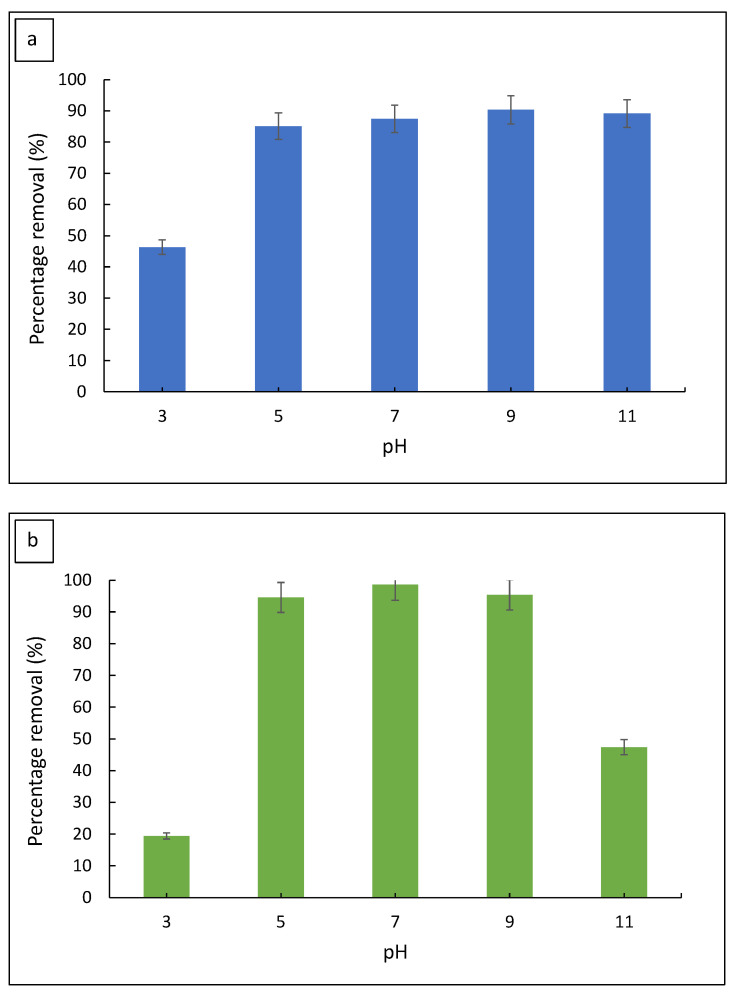
The effect of the initial dye solution pH on the percentage removal of (**a**) MB and (**b**) MG.

**Figure 6 polymers-14-05416-f006:**
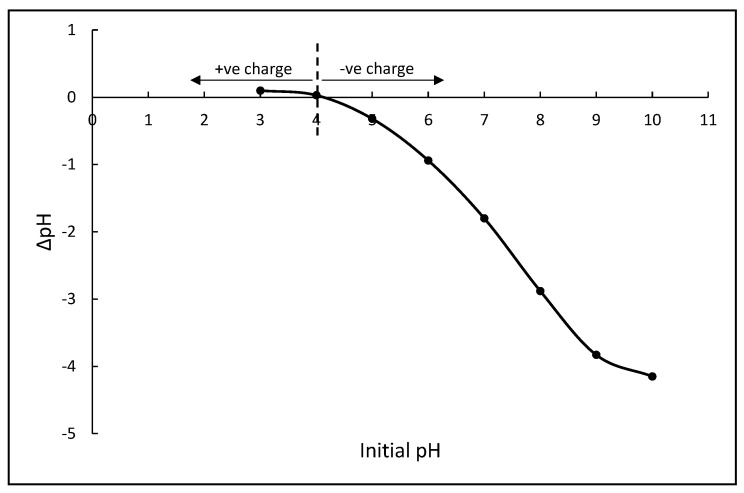
The zero−point charge of FMS/PU.

**Figure 7 polymers-14-05416-f007:**
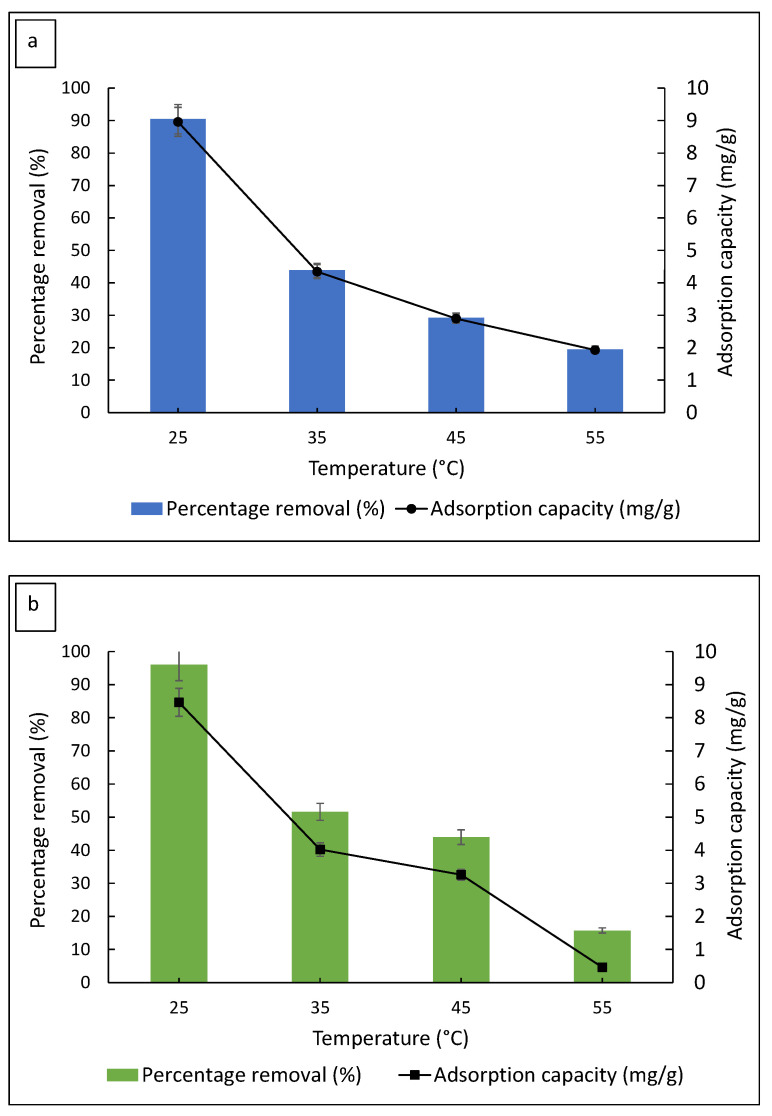
The effect of the solution temperature on the percentage removal of (**a**) MB and (**b**) MG.

**Figure 8 polymers-14-05416-f008:**
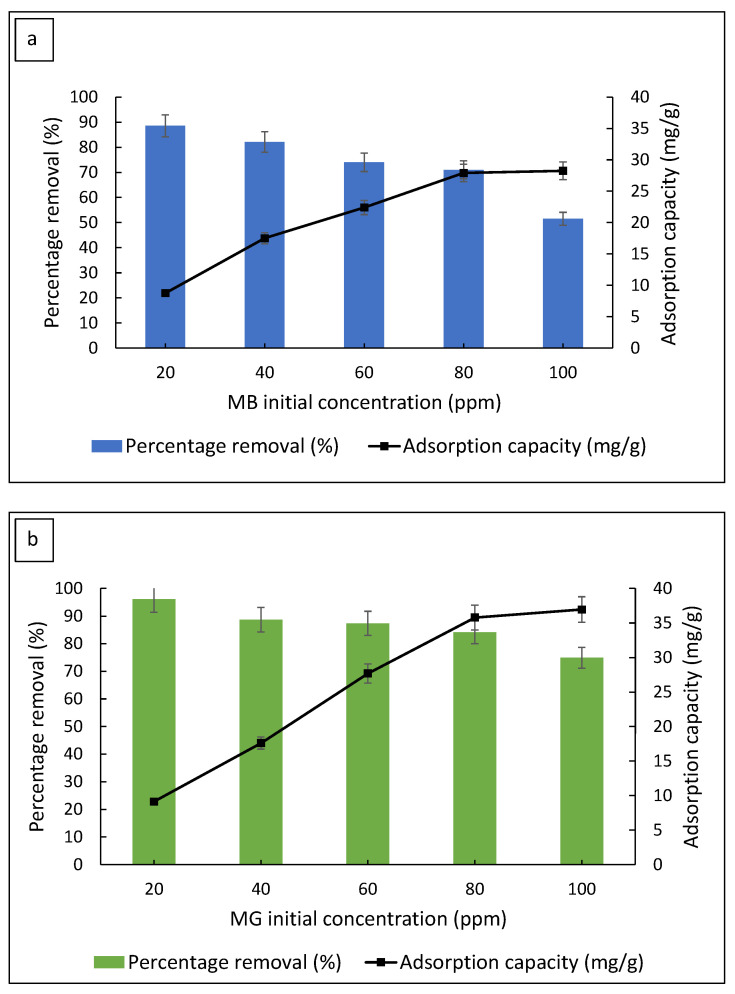
The effect of the initial dye concentration on the percentage removal and adsorption capacity for (**a**) MB and (**b**) MG.

**Figure 9 polymers-14-05416-f009:**
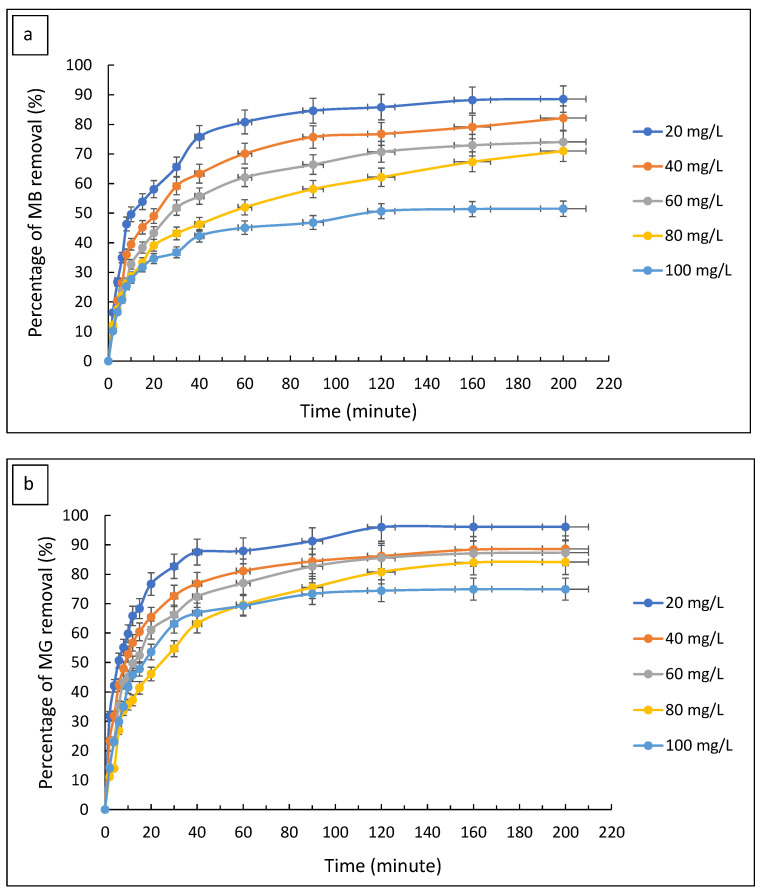
The effect of the adsorption time on the percentage removal of (**a**) MB and (**b**) MG.

**Figure 10 polymers-14-05416-f010:**
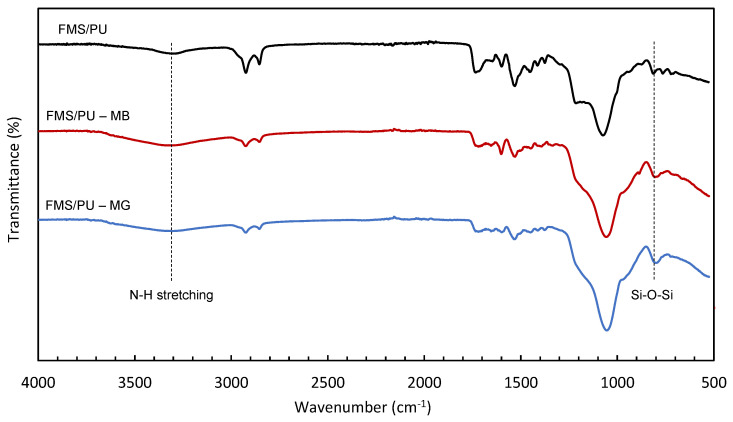
FTIR spectra of FMS/PU, FMS/PU−MB, and FMS/PU−MG.

**Figure 11 polymers-14-05416-f011:**
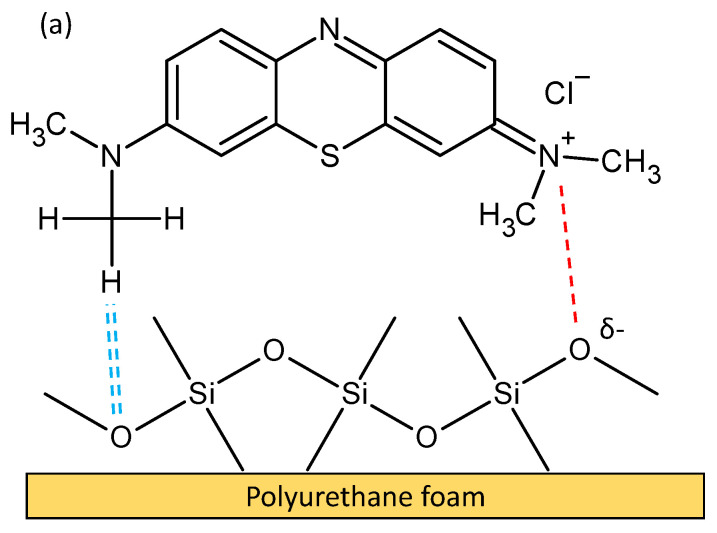
The proposed adsorption mechanism of (**a**) MB and (**b**) MG onto FMS/PU.

**Figure 12 polymers-14-05416-f012:**
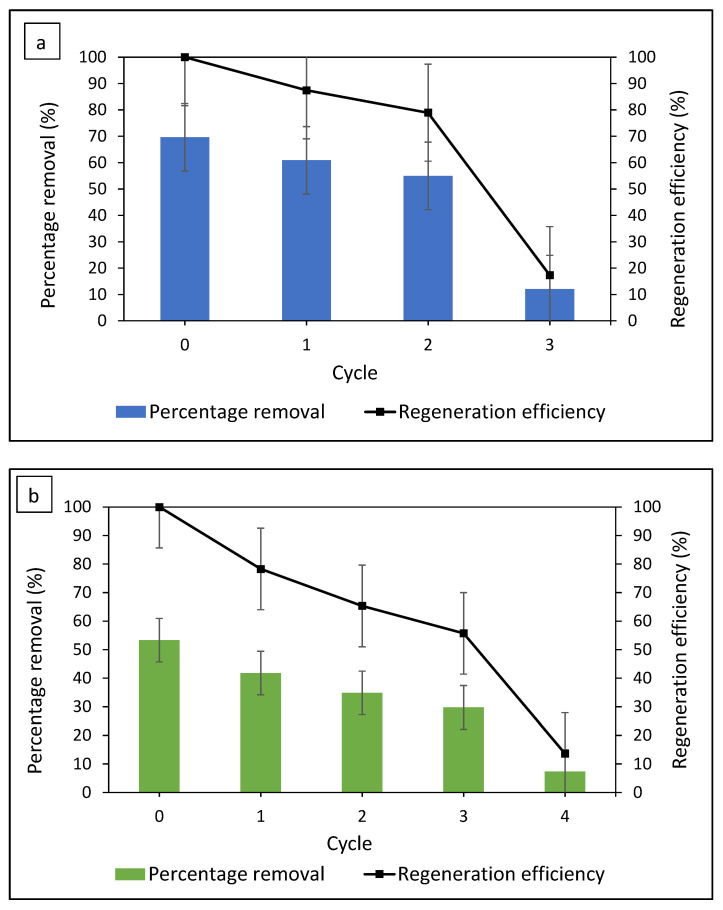
The regeneration cycle of the FMS/PU composite for (**a**) MB and (**b**) MG.

**Figure 13 polymers-14-05416-f013:**
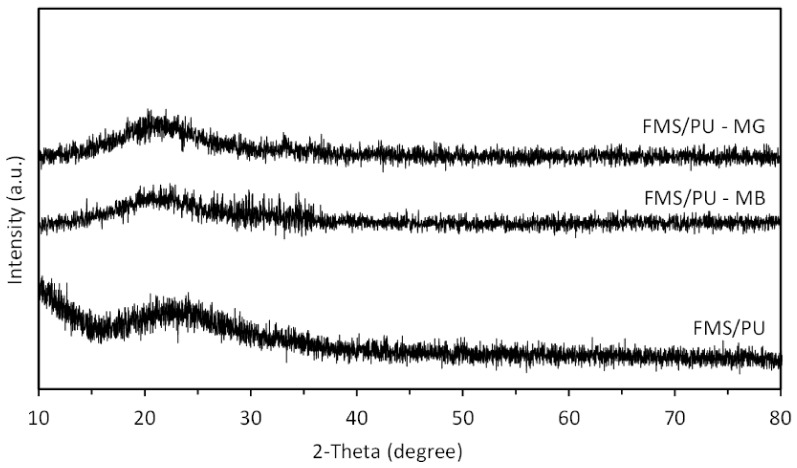
X-ray diffraction of FMS/PU, FMS/PU—MB, and FMS/PU—MG.

**Figure 14 polymers-14-05416-f014:**
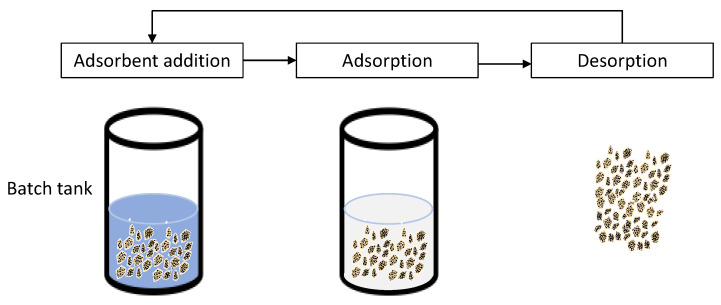
Typical setup for pollutant removal via batch adsorption [64].

**Table 1 polymers-14-05416-t001:** Parameters for the pseudo-first order, pseudo-second order, Elovich kinetic model, and intraparticle diffusion for MB.

Dye	MB				
Concentration	20	40	60	80	100
Pseudo First Order					
*Q_e_* experimental (mg/g)	8.8	17.5	22.4	27.9	28.3
*Q_e_* calculated (mg/g)	6.3	14.1	18.0	19.7	21.8
*k* _1_	0.0835	0.0835	0.0772	0.0778	0.0575
*R* ^2^	0.8953	0.9388	0.9634	0.8734	0.9371
Pseudo Second Order					
*Q_e_* experimental (mg/g)	8.8	17.5	22.4	27.9	28.3
*Q_e_* calculated (mg/g)	7.8	16.4	19.3	20.2	23.8
*k* _2_	73.5	452.8	872.5	1297.0	2592.0
*R* ^2^	0.9859	0.9663	0.9834	0.9919	0.9949
Elovich Kinetic Model					
*α* (mg/(g.min))	3.602	4.866	0.816	8.551	14.448
*β* (g/mg)	0.6799	0.3254	0.2548	0.2789	0.2276
*R* ^2^	0.9599	0.9390	0.9534	0.9353	0.9556
Intraparticle Diffusion					
*k_IP_* (mg/(g.min^1/2^))	0.983	2.094	2.676	2.641	2.933
*C_IP_* (mg/g)	1.305	1.249	1.120	2.870	5.207
*R* ^2^	0.9641	0.9796	0.9956	0.9846	0.9571

**Table 2 polymers-14-05416-t002:** Parameters for pseudo-first order, pseudo-second order, Elovich kinetic model, and intraparticle diffusion for MG.

Dye	MG				
Concentration	20	40	60	80	100
Pseudo-First Order					
*Q_e_* experimental (mg/g)	9.1	17.6	27.7	35.8	38.0
*Q_e_* calculated (mg/g)	5.8	11.7	20.7	29.9	28.9
*k* _1_	0.1211	0.1054	0.0941	0.1229	0.0660
*R* ^2^	0.9671	0.9476	0.9317	0.9721	0.9348
Pseudo-Second Order					
*Q_e_* experimental (mg/g)	9.1	17.6	27.7	35.8	38.0
*Q_e_* calculated (mg/g)	8.8	16.4	26.1	29.0	39.4
*k* _2_	166.8	936.7	2414.0	2909.1	7680.9
*R* ^2^	0.9956	0.9961	0.9827	0.9766	0.9846
Elovich Kinetic Model					
*α* (mg/(g.min))	8.553	10.198	9.264	7.275	13.495
*β* (g/mg)	0.6768	0.3268	0.1935	0.1585	0.1324
*R* ^2^	0.9477	0.9765	0.9736	0.9889	0.9419
Intraparticle Diffusion					
*k_IP_* (mg/(g.min^1/2^))	1.006	2.203	3.466	4.205	5.163
*C_IP_* (mg/g)	2.518	3.640	2.935	0.892	3.9256
*R* ^2^	0.9871	0.9856	0.9838	0.9870	0.9890

**Table 3 polymers-14-05416-t003:** Parameters for the Langmuir isotherm, Freundlich isotherm, and separation factor for MB and MG adsorption.

	Langmuir Isotherm			Freundlich Isotherm				Separation Factor
Dye	*k_L_*	*Q_max_*	*R* ^2^	Q_max_	*k_F_*	1/*n*	*R* ^2^	*R_L_*
MB	0.1761	31.7	0.9954	42.9	7.323	0.3838	0.9189	0.054–0.221
MG	0.4875	34.3	0.9439	76.1	10.49	0.4305	0.9591	0.020–0.093

**Table 4 polymers-14-05416-t004:** Comparison of different silica-containing adsorbents used to remove MB and MG.

	Adsorbent	*Q_max_* (mg/g)	Reference
MB	FMS/PU composite	31.7	*This work*
	Silica hollow nanosphere-3	25.5	Liu et al. [55]
	Silica nanosheet	12.6	Zhao et al. [56]
	CMC Fe_3_O_4_@SiO_2_ MNP	22.7	Zirak et al. [12]
	Fe_3_O_4_@SiO_2_-CR	31.4	Yimin et al. [57]
MG	FMS/PU composite	34.3	*This work*
	Xerogel activated diatom (XDE)	4.2	Sriram et al. [58]
	Silica aerogel (HSA)	6.7	Liu et al. [59]
	Fe_3_O_4_@SiO_2_-CPTS	12.5	Feyzi et al. [60]
	Diatomite	23.6	Tian et al. [61]

**Table 5 polymers-14-05416-t005:** The thermodynamic parameters for MB and MG adsorption.

Dye	Temperature (K)	*K_d_*	Parameters		
			ΔG° (kJ/mol)	ΔH° (kJ/mol)	ΔS° (J/mol K)
MB	298	4.7194	−2.55	−95.43	−311.69
	308	0.3903	0.56		
	318	0.2068	3.68		
	328	0.1210	6.79		
MG	298	12.0977	−5.09	−151.48	−491.24
	308	0.4189	−0.18		
	318	0.2935	4.72		
	328	0.0277	9.63		

## Data Availability

All data used during the study are provided in the manuscript.

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
