# Peer review of "Removal of Cationic Dyes by Iron Modified Silica/Polyurethane Composite: Kinetic, Isotherm and Thermodynamic Analyses, and Regeneration via Advanced Oxidation Process"

_polymers, 2022, doi:10.3390/polym14245416_

Round 1
Reviewer 1 Report
Authors herein reported the synthesis of IMS/PU adsorbent and investigated its performance for removing cationic dyes and the regeneration using AOP technology. This manuscript can be accepted but the below issues should be addressed.
1. More technologies are needed to characterize the physicochemical properties of their materials, such as SEM for morphologic observation, and FTIR for confirmation of functional groups.
2. The adsorption behaviors of their materials should be compared with the reported adsorbents in references.
3. They should discuss the possible reason for the drastic decrease in adsorption amount of the 4th run.
4. FTIR of the regenerated sample is suggested to confirm the removal of the adsorbed dyes.
Reviewer 2 Report
This manuscript reported the preparation of the iron modified silica/polyurethane (FMS/PU) composite. As the adsorbents, the effects of various parameters such as adsorbent load, pH, initial dye concentration and contact time on the removal of cationic dyes (MB and MG) were demonstrated. Adsorption mechanism and recycle performances were also discussed. The obtained results were of certain interests for related readers. Therefore, I recommend it to be published.
Some issues should be addressed, as follows:
Q1. In equation 1, what does “x” mean?
Q2. In the characterization section, UV-vis and XRD models should be provided. Particularly, the operated voltage and current in XRD measurement should be given.
Q3. Many formatting problems. For example, k and R2 in Table 1 should be in italics.
Q4. The MG in Figure 10 was different from the MG in Figure 1.
Q5. The references in the References section should be typeset according to the Polymers requirements.
Q6. Why did MB and MG were chosen as two cationic dyes? What was the relationship between the two? Can you explain that adsorption MB was better than adsorption MG on the basis of the proposed mechanism?
